# Predicting the Structure of Enzymes with Metal Cofactors: The Example of [FeFe] Hydrogenases

**DOI:** 10.3390/ijms25073663

**Published:** 2024-03-25

**Authors:** Simone Botticelli, Giovanni La Penna, Velia Minicozzi, Francesco Stellato, Silvia Morante, Giancarlo Rossi, Cecilia Faraloni

**Affiliations:** 1Department of Physics, University of Roma Tor Vergata, 00133 Rome, Italy; simone.botticelli@roma2.infn.i (S.B.); velia.minicozzi@roma2.infn.it (V.M.); francesco.stellato@roma2.infn.it (F.S.); silvia.morante@roma2.infn.it (S.M.); rossig@roma2.infn.it (G.R.); 2Section of Roma Tor Vergata, National Institute of Nuclear Physics, 00133 Rome, Italy; 3Institute of Chemistry of Organometallic Compounds, National Research Council, 50019 Florence, Italy; 4Museo Storico della Fisica e Centro Studi e Ricerche E. Fermi, 00184 Rome, Italy; 5Institute of Bioeconomy, National Research Council, 50019 Florence, Italy

**Keywords:** hydrogenase, molecular modelling, structure prediction, microalgae, photobiological hydrogen production

## Abstract

The advent of deep learning algorithms for protein folding opened a new era in the ability of predicting and optimizing the function of proteins once the sequence is known. The task is more intricate when cofactors like metal ions or small ligands are essential to functioning. In this case, the combined use of traditional simulation methods based on interatomic force fields and deep learning predictions is mandatory. We use the example of [FeFe] hydrogenases, enzymes of unicellular algae promising for biotechnology applications to illustrate this situation. [FeFe] hydrogenase is an iron–sulfur protein that catalyzes the chemical reduction of protons dissolved in liquid water into molecular hydrogen as a gas. Hydrogen production efficiency and cell sensitivity to dioxygen are important parameters to optimize the industrial applications of biological hydrogen production. Both parameters are related to the organization of iron–sulfur clusters within protein domains. In this work, we propose possible three-dimensional structures of *Chlorella vulgaris* 211/11P [FeFe] hydrogenase, the sequence of which was extracted from the recently published genome of the given strain. Initial structural models are built using: (i) the deep learning algorithm AlphaFold; (ii) the homology modeling server SwissModel; (iii) a manual construction based on the best known bacterial crystal structure. Missing iron–sulfur clusters are included and microsecond-long molecular dynamics of initial structures embedded into the water solution environment were performed. Multiple-walkers metadynamics was also used to enhance the sampling of structures encompassing both functional and non-functional organizations of iron–sulfur clusters. The resulting structural model provided by deep learning is consistent with functional [FeFe] hydrogenase characterized by peculiar interactions between cofactors and the protein matrix.

## 1. Introduction

The enzyme [FeFe] hydrogenase (Hyd, hereafter) is found in bacteria and microalgae organisms [1]. The enzyme is an efficient catalyst in converting the proton of liquid water into molecular hydrogen as a gas, using either solar energy or chemical reductants [2,3]. Among different types of hydrogenases, the [FeFe] hydrogenase is more efficient in hydrogen production than the other types. Therefore, Hyd is at the core of biological production of H_2_, which is a cost-effective and carbon-free strategy to produce fuels [4]. Among the organisms that express the enzyme, *Chlorella vulgaris* 211/11P *Cvu*, is an interesting microalga strain that presents a promising candidate for the production of biofuels at an industrial scale [5], due to its rapid growth in fresh water. Strain selection is fundamental to optimize the biological production of H_2_ by microalgae photosynthesis at a reasonable cost [6]. The proton reduction takes place in the so-called H-cluster of Hyd, which is the active site inside the evolutionary preserved core domain of [FeFe] hydrogenases. *Cvu* microalgae were found to produce H_2_ gas in relatively easy growing conditions [7]. H-cluster is arranged in a canonical cubane-type iron–sulfur cluster, [4Fe4S]_*H*_, which is linked via the thiol group of a cysteine amino acid to a peculiar diiron active site, [2Fe]_*H*_. The two iron atoms of [2Fe]_*H*_, in proximal (Fe_*p*_) or distal (Fe_*d*_) positions relative to [4Fe4S]_*H*_, carry biologically atypical carbon monoxide (CO) and cyanide (CN^−^) ligands that stabilize the reduced Fe states. The latter ligand molecules are present in all classes of hydrogenases [8]. The two ligand molecules azadithiolate (−(SCH_2_)_2_NH, *adt* hereafter), and one of the CO molecules bridge Fe_*p*_ and Fe_*d*_ in [2Fe]_*H*_, respectively. The catalytic mechanism likely involves the formation of an intermediate hydride species of the H-cluster [9]. In Figure 1, we display the structure of the Hyd H-cluster of the bacterium *Clostridium pasteurianum* [FeFe] hydrogenase I (*CpI*, hereafter), together with a schematic drawing of its chemical structure.

While the role of the protein domain where the H-cluster is embedded (the H-domain) has been investigated in detail, the structural and functional role of other [FeFe] hydrogenase regions are more elusive, because limited structural information is available. Most [FeFe] hydrogenases possess additional FeS clusters, called F-clusters. The task of these auxiliary (or accessory) clusters is to transfer electrons from external donors to the catalytic site and vice versa. In the case of [FeFe] hydrogenase, where the electron external donor is often reduced ferredoxin, auxiliary clusters allow electron transfer from the protein surface where ferredoxin binds to the active site. The latter is buried into the H-domain to attract electrons, favouring protection of active reduced states from oxidation. Because of the larger solvent accessibility of F-clusters compared to the buried H-cluster, the properties of the former may influence the O_2_ sensitivity more significantly than the latter, as O_2_ is dissolved in the solution environment. Since the H-cluster environment is very conserved within all [FeFe] hydrogenases, it is reasonable to assume that the auxiliary clusters may differently tune O_2_ sensitivity in different species [11]. From this point of view, the eventual structure of the F-domain and its interactions with the H-domain can be important features to possibly increase the resistance of Hyd to O_2_ [9].

For completeness, the three-dimensional (3D) structures of three [FeFe] hydrogenases obtained by X-ray diffraction of crystals are displayed in Figure 2. The location of the H-clusters and the auxiliary [4Fe4S] F-clusters ([4Fe4S]_*F*_, hereafter) are explicitly indicated.

The first two structures refer to the bacteria *Clostridium pasteurianum* and *Desulfovibrio desulfuricans* (*Dd*, hereafter), respectively. They are the best characterized enzymes in the literature in terms of structure [2]. These hydrogenases are, depending on metabolites, location, and other cellular conditions, catalysts for either hydrogen uptake or hydrogen production [16]. The *Chlamydomonas rheinardtii* (*Cr*) Hyd structure is the only [FeFe] hydrogenase from microalgae, which has been crystallized, though the crystal lacks the [2Fe]_*H*_ subcluster. This hydrogenase enzyme is one of the most efficient catalysts for hydrogen gas production at (temperature and pressure) room conditions. Most of the other *Cr* Hyd structures in the protein databases (see Appendix A ) are almost identical to 3LX4. Even the O_2_ inactivated HydA1 enzyme (PDB 4R0V [17]) has a structural deviation (measured as RMSD) from 3LX4 of 0.7 Å. The overall structure of the H-domain in *Cr* Hyd is similar to that in *CpI* and *Dd* Hyd. The most interesting difference between *Cr* Hyd and the bacterial Hyd is the absence of a folded F-domain including some auxiliary clusters. *Cr* Hyd is considered as the minimal catalyst of biological H_2_ production, though very sensitive to oxidation. Compared to genus *Chlamydomonas*, genus *Chlorella* microalgae can have folded F-domain and auxiliary clusters efficiently bound [11], but no crystal structures are available.

The isolation and purification of [FeFe] hydrogenase from algae is problematic [18] and this is one of the reasons why in vitro characterization of Hyd in species different from bacteria is missing, with a few exceptions though. Hyd is part of the dioxygen sensing mechanism developed by unicellular eukaryotes to adapt their metabolism to environments where the production of dioxygen by photosynthesis is not fully sustainable [16].

In this investigation, we perform the first computational study proposing a structural and functional characterization of hydrogenase in *Cvu* where also the FeS clusters required for hydrogenase functioning are included. To predict possible structures for the protein including the location and binding of iron–sulfur clusters, we start from the protein sequence employing the first two proposals produced by a routinely used tool to predict draft structures, SwissModel [19]. The first proposal is obtained by the deep learning algorithm, AlphaFold [20]. The prediction pipeline contains different moduli encompassing, in the first steps, many of the established homology modeling techniques, but in later steps builds configurations of disordered regions by using conformational maps. This structure was reproduced by us using the original AlphaFold code. The second structure is obtained by homology method (from SwissModel) based on the search of HHblits [21]. This method ignores conformational maps and therefore proposes configurations of disordered regions on the basis of the available experimental structures. Thus, a number of regions cannot be predicted and are consequently missing. Deep-learning and homology methods have been compared [22], concluding that in the absence of sufficient experimental information reliability of results depends on the specific application [23]. The two above mentioned structures are compared to a third manually constructed model, based on the highest resolution crystal structure available for [FeFe] hydrogenase. Interatomic potentials are used to set the initial structures in the environment similar to chloroplast stroma where microalgae hydrogenase is located [24].

Summarizing, we propose structures for *Cvu* Hyd according to the following protocol: (i) we analyze the annotation of the sequence as available in the literature; (ii) we assume the existence of a relation between the protein matrix and two iron–sulfur clusters, the catalytic one in the H-domain and a second accessory one in a hypothetical F-domain; (iii) we use deep-learning and homology methods to predict 3D initial structures; (iv) we perform simulations of a third molecular models where the protein, the two iron–sulfur cofactors, and a simple model of water solution environment are included.

This work extends that performed by Ordonez et al. [25], in that here we fully annotate the gene in *Cvu* which potentially expresses Hyd and we provide a reproducible pipeline designed to obtain 3D structures including functional ones. The annotated *Cvu* sequence is different from any other one previously proposed in computational modeling. The knowledge of the enzyme structure is necessary for further mechanistic studies [12]: isolation and purification of [FeFe] hydrogenase from microalgae, with yields allowing structural in vitro studies, are very difficult [18]. Because of enzyme isolation issues, even the *Cr* enzyme could be characterized in various states only in whole E. coli cells [26,27].

Our construction provides the basis for interpreting possible forthcoming experimental data coming from a possible new *Cvu* hydrogenase endowed with a single potential F-cluster on top of the H-cluster.

## 2. Results

### 2.1. Alignment and Gene Annotation

The *Cvu* Hyd sequence was determined by using the procedure described below in Methods (Section 4). In performing the alignment of the whole set of (10,723) *Cvu* nuclear genes to the eight known Hyd sequences (see Figure 3), only one gene displays both E-value smaller than 3 × $10^{-19}$ and query coverage larger than 58%. The gene in question is KAI3438965.1. It is interesting to note that for this gene the UNIPROT [28] data-base yields the following annotation: “Iron hydrogenase small subunit domain-containing protein”.

The alignments reveal in all cases the potential fingerprint of the H-cluster binding domains composed by 3 conserved motifs [16,29].

Although in Ref. [29] only the most studied Hyd are shown and compared, the full list of well-conserved motifs is available and can be found in Refs. [16,30]. There are evident changes to sequences that have been identified in P1-3 motifs in the literature. In particular, the TSC sequence in the P1 motif of *Cvu* Hyd is not present in the known Hyd sequences. The P1 sequence change is, however, also observed in one *Cv* gene (XP_005843810.1), that is the protein sequence closest to that analyzed in this work and that bears the same annotation. The most important residues are the conserved Cys 224, 280, 465, 469 in *Cvu* Hyd sequence (red in Figure 3), all binding the H-cluster. Apart from not fully conserved residues, the alignments we have performed show that the sequence extracted from the whole *Cvu* genome represents, especially in terms of the position of Cys residues, a candidate for the FeS cluster ligand, as it is required by [FeFe] hydrogenase. In Ref. [16] (Figure 12) and first line in Figure 3 the second Cys in P1 is not always present, while it is a marker in L1 [29].

By using the available hydrogenase identifier tool [31], the identified *Cvu* gene is not classified as a hydrogenase and the same is true for the similar sequence of *Cv* (XP_005843810.1). Conversely, the *Cv* gene AEA34989.1 (see below) is identified as hydrogenase of class A. It must be noticed that in the sequence alignment of all the *Cvu* genes, no other genes display similarity with any of the existing hydrogenases, including gene AEA34989.1 of *Cv*. Since both species produce hydrogen, the whole genome search excludes that other genes can code for a possible hydrogenase in *Cvu*, differently from *Cv*.

It is interesting to notice the fairly good alignment, in fingerprint regions, between the KAI3438965.1 *Cvu* gene and the genes expressing the nuclear prelamin A recognition factor (Nar) in eukaryotic organisms [32] (see the six bottom lines in Figure 3). The Nar class of proteins has been found as a possible evidence of a “memory” of ancient Hyd in aerobic eukaryotes [16]. The sequence variability between the Hyd and Nar classes is larger in the direction of N-terminus: P1 looks more variable than P3 and indeed only Nar proteins show the ASA sequence in place of TSC. The implications of the similarity of the addressed *Cvu* gene to Nar genes concerning the ability of binding H- and F-clusters are described in more details in Section 3.

Since there are no other genes among the *Cvu* candidates for H-cluster binding while the hydrogenase activity was demonstrated in *Cvu* cultures, we are led to conclude that the hydrogenase sequence of *Cvu* is gene KAI3438965.1. Naturally, we cannot exclude that the expressed Hyd protein is eventually processed when introduced in chloroplasts and there assembled into the [2Fe]_*H*_ component of H-cluster.

By inspecting the global alignments (see in Appendix A, the folder Blas_Clustal_Alignments) we observe that also the Cys residues in the N-termini of *Cvu* Hyd are aligned to the species where Cys residues are present in the N-terminal region, namely *CpI*, *Dd* Hyd, *Cv* Hyd. In Figure 4, we display, as an example, the alignment between *Cvu* and *Cv* Hyd (gene AEA34989.1), the latter belonging to class A Hyd [31] (see above). Previous studies about the recruitment of auxiliary F-clusters along with evolutionary analysis arguments showed that Hyd in species without these aligned Cys residues in the N-terminal region (like *Cr*, *Cf*, *CspDT*, and *Td*, does not bind auxiliary F-clusters [29]. Therefore, our alignment, that is common to *CpI* and *Dd* Hyd, shows that an auxiliary conserved F-cluster can be present in *Cvu* Hyd as well as in the other Hyd sequences where an F-domain has been identified (see the crystal structures of *CpI* and *Dd* Hyd).

In the following analysis, we assign Cys 72, 75 and 78 in *Cvu* Hyd as ligands for an auxiliary F-cluster. To complete the binding of the auxiliary 4Fe4S cluster, we assign Cys 21 as a fourth ligand. In Nar proteins, the 4Fe4S binding pattern by four Cys residues is found to be Cx_38_Cx_2_Cx_2_C [33], while in *Cvu* we find Cx_50_Cx_2_Cx_2_C (x denotes a generic aminoacid). As shown by Nar proteins, the actual cooperation between the first Cys residue and the other three Cys residues depends on the structure of the long protein segment (38 residues) connecting the two regions. The 4-Cys binding of the auxiliary F-cluster is here adopted on the basis of the homology with the Nar proteins. The assessment of its stability is the major aim of the following study.

In summary, by using the sequence alignment of the KAI3438965.1 *Cvu* gene discussed above, we observe that the [FeFe] hydrogenase sequences feature the following three domains: the N-terminal F-domain, with at least one possible F-cluster; the central well-structured and conserved H-domain, containing the active site (H-cluster) bound by Cys residues in P1, P2, and P3 motifs; an unstructured C-terminal short segment. These domains are characterized in terms of residue numbers in Table 1.

In Table 1, we report the location along the amino acid sequence of the Cys residues that bind the FeS clusters in the *Cvu* F- and H-domain and C-terminus.

To confirm evidence of the *Cvu* gene KAI3438965.1 annotation to a hydrogenase function, we have performed a detailed analysis of the structural protein stability as described in the following section.

### 2.2. Structure Prediction

As is commonly done in similar cases, we inserted the KAI3438965.1 sequence in the SwissModel server (see Section 4 for details) and decided to continue the sequence structural study taking into further consideration the first two of the proposed structures produced in this search.

To reproduce the 3D structure of the first proposal of *Cvu* Hyd obtained by SwissModel, we used the deep learning algorithm (AlphaFold, AF hereafter, see Section 4).

Before going into the details of the 3D AF predictions, it is convenient to first describe the distribution of structural information among the sequence domains detailed in the previous section. In all available Hyd crystal structures, namely (*CpI* 6N59, *Dd* 1HFE and *Cr* 3LX4), the H-domain scaffold is made by 11 α-helical secondary domains embedded into a globule with a further small β-sheet made of 7 β strands in an antiparallel configuration. We call this arrangement of secondary motifs the “scaffold” of the H-domain. In Appendix A, we indicate the residues belonging to the scaffold shared by *Cvu* Hyd and the three reference available crystal structures mentioned above.

We first notice that the formation of disulfide bridges between Cys residues has never been observed in hydrogenase structures obtained in anaerobic in vitro working conditions. This is confirmed by the AF predictions with no surprise as they are obtained from the available structural information. In the chloroplast, stroma hydrogenase is functional when in a reducing environment, while in the presence of dioxygen it is rapidly degraded. Thus, in the following structural refinement we shall assume that no disulfide bridges are present in the models in study.

In the AF output file “pdb_hits.hrr” (see Appendix A AF_alignments/msas directory), secondary structure motifs of crystal structures coming from database search and AF predicted structures are compared using the DSSP algorithm [34]. AF then calculates the predicted alignment error (PAE) matrix [35]. The matrix provides AF’s expected position error between pairs residues and it is a measure of the confidence in the relative position of two residues within the predicted structure. As displayed in Appendix A, PAE matrix for *Cvu* prediction shows large errors for three inter-domain, the hypotetical F-domain and the two extended loops visible in Appendix A (left panel). This fact indicates that the relative positions and/or orientations of at least two segments connecting two ordered regions in the 3D structure are poorly determined. Therefore, F-domain and the disordered segments must be refined by, for instance, molecular simulations to reduce this uncertainty.

In Appendix A, we display the behavior of the structural prediction (plDDT) accuracy index used by AF along with the protein sequence in the case of the highest-score AF structure prediction (see Section 4 for details). The other predicted structures do not differ significantly in the H-domain secondary structure. The structural difference is limited to the same regions identified by PAE, in particular in the terminal regions, i.e., the N-terminus that is the possible F-domain, and the extended loops connecting secondary domains lying within the H-domain (see Appendix A, left panel).

The scaffold of the H-domain (displayed as top-right horizontal bars in the figure) is entirely made by residues with high plDDT index (blue points). The long unstructured loops that connect secondary motifs within the H-domain (residues Gly 173-Gly 212 and Ala 338-Ser 381) are constructed by AF uniquely to connect regions where structural information is high. On the other hand, the F-domain is characterized by a partially emerging secondary structure. Indeed, a short α-helix (residues Ser 81-His 89) is found by the STRIDE algorithm at the interface between the F- and the H-domain (Appendix A, left panel). Interestingly, an apparently rigid turn encompasses residues Leu 69-Asp 71 with the consequence that Cys 72, 75, 78 become close in space because of the constraint imposed by these secondary motifs. This information comes from the available crystal structures that AF elaborates, taking into account the presence of the auxiliary cluster in *CpI* that is proximal to its H-cluster. In order to better quantify the deviation of the structural prediction with respect to known available 3D structures (three crystal structures and one recent electron microscopy based model), in Table 2 we display the root-mean square deviation (RMSD, see Methods) of heavy backbone atoms belonging to different sets of aligned residues.

We measured the structural deviation among two portions of the scaffold: the 11-helices bundle and the 7 β-strands. In all cases, the deviation is measured by comparing the *Cvu* AF prediction with three reference known crystal structures and one recent cryo electron-microscopy (cEM) structure. These are *CpI*, *Dd* Hyd, *Cr* Hyd, and *Tm* HydABC. The latter is one of the chains (PDB 7P92) reported for the *Thermotoga maritima* HydABC electron bifurcating [FeFe] hydrogenase, indicated as HydABC [36]. The lower values of RMSD obtained for the β-strands compared to the values coming from the α-helices points to a more rigid structure of the H-domain portion formed by the seven β-strands. This analysis suggests that the H-domain, independently of the hydrogenase performance, can be divided in two sub-domains: one made by the α helices; the other by the two β sheets. The two domains will be used as references to describe the equilibration of the *Cvu* Hyd model into the solution environment modeled in the following simulations. As expected, after energy minimization (see below), AF predictions of the H-domain structure differ mainly in the two large extended loops.

In the following, we introduce labels that will be used in the subsequent structure refinement. Notice that this step is performed once the FeS clusters are explicitly added to initial structural models and atomistic simulations are performed. We denoted the simulation starting from the AF construction described above as simulation **1**.

A second simulation (**2**, hereafter) was started from the F-domain proposed by AF combined with the H-domain proposed by SwissModel. The H-domain is present, in the SwissModel second proposal, in the form of a partially oxidized HydA1 protein of *Cr* (PDB 4R0V [17]). This initial configuration displays the following RMSD values (in Å see Table 2) with respect to the sequence in parenthesis: 4.8/4.2/5.8 (*CpI*); 4.7/4.1/5.8 (*Dd*); 3.7/2.3/5.7 (*Cr*); 5.0/4.5/5.6 (*Tm*); 22.8/5.6/6.0 (*Cvu* **1**). It can be noticed that the deviation of the entire scaffold is smaller than in AlphaFold proposal, while the deviation of the β-sheet is larger. Indeed, the first β-strand (b1, see Appendix A) does not get formed at the beginning of simulation **2**.

The further simulation where the F-domain has the initial backbone structure of *CpI* was denoted as simulation **3** (see Section 4 for details). In this case, the initial RMSD (see Table 2) with respect to the reference known structures is the same as in **1**.

### 2.3. Structure Refinement

In view of the time evolution of several structural parameters like gyration radius, solvent-accessible surface area (SASA), distance between FeS clusters (see Appendix A and related discussion in Appendix A), observed in molecular dynamics (MD) we decided to compare structural parameters averaging over the last 200 ns of the simulated MD trajectories. Simulations show that the initial configurations require several hundreds ns to settle down in the water environment. Reduction of fluctuations and the stabilization of the monitored structural parameters after 800 ns of simulation are the result of the equilibration of different initial configurations of the system.

The small structural deviation of FeS clusters in all simulations indicates that they all keep a structure functional to hydrogenase activity (see Appendix A). One A crucial step underlying the hydrogenase activity is the proton reduction occurring at the Fe_*d*_ site. To probe the proton access to this site, we measured the accessibility of solvent water molecules to Fe atoms. The radial distribution function, g(r) (Figure 5) of Fe-Ow pairs shows that the distal Fe_*d*_ atom is accessible to water in simulations **1** and **3**, while Fe_*p*_ is less accessible than Fe_*d*_ in all simulations (left panels). This is a consequence of the free valence left on Fe_*d*_ in all models. However, the more compact H-domain used as initial configuration in simulation **2** displays a less accessible Fe_*d*_ atom, because of the relatively smaller number of water molecules in the Fe first-coordination shell (r< 0.4 nm). Fe atoms of [4Fe4S]_*H*_ are buried, while Fe atoms in [4Fe4S]_*F*_ are more accessible to water molecules (right panels). Furthermore, Fe atoms in [4Fe4S]_*H*_ clusters are significantly more accessible in the first coordination sphere in simulation **2** compared to **1** and **3**. The water molecules accessing Fe_*d*_ are exchanged by water molecules at larger distances, showing that water molecules are not trapped into protein cages because of the bias induced by the initial model construction.

Interestingly, in all simulations we found that the H atoms of water molecules approaching Fe_*d*_ are located at a distance smaller (2.5 Å) than the corresponding O atom (3.5 Å, see the leftmost density in g(r) in Figure 5). The probabilities of such water configuration are 0.07, 0.04, and 0.11 for simulations **1**, **2** and **3**, respectively. This means that the water molecule approaching Fe_*d*_ is often oriented with an H atom towards Fe_*d*_, the site of H reduction. In Figure 6, we show a representative configuration of the F- and H-clusters after a simulation time of about 800 ns of model **1**. Figure 6 shows that the water molecule approaching Fe_*d*_ atom has the orientation expected for an efficient proton reduction [37]. In fact, one proton is at 2.1 Å from Fe_*d*_ and forms a hydrogen bond with N(*adt*) (2.2 Å). The proton of NH(*adt*) is also at a pretty small distance from Fe_*d*_, namely 2.2 ± 0.5 Å in (**1**) and 2.5 ± 0.6 Å in (**2**).

The average distances between the [4Fe4S]_*F*_ and [4Fe4S]_*H*_ cluster centers of mass (see also Appendix A are: 17.3±0.6 Å in (**1**); 34.9±0.8 Å in (**2**); 15.1±0.9 Å in (**3**). In simulations **1** and **2**, the average distance is only slightly above the upper range of distances where electron transfer between the two clusters can occur in proteins [38,39]. However, we should consider this distance remarkably small for simulations **1** and **2** considering the fact that we started from an extended configuration of the F-domain produced by AF and SwissModel, respectively. On the other hand, the large distance observed in simulation **3** points to structural constraints in the F-domain that hinder the approach of the two clusters (see below). As discussed in a recent work [37], active [FeFe] hydrogenases, with the exception of *Cr*, show that the [4Fe4S]_*H*_-[4Fe4S]_*F*_ distance (measured as the distance between the centers of mass) is in the range 11–14 Å. The comparison of simulations shows that simulations **1** and **2** are in better agreement with the above observations.

The difference between the simulations is to be ascribed to different model constructions as most of the changes in distance develop during the first 300 ns (see the time evolution of the structural parameters in Appendix A).

The above observations regarding the simulation **1** can be summazized by saying that the collected simulation data are consistent with: (i) the mechanism of proton reduction at Fe_*d*_; (ii) the protection to oxidation of [4Fe4S]_*H*_; (iii) a good accessibility of the F-cluster which interacts with reductant species coming from the protein environment.

In the following, we shall describe interactions relevant to stabilize reactive configurations.

In *Cvu* Hyd the first three residues of the protein motif facing Fe_*d*_ (motif P1, see Figure 3) are not conserved among the other [FeFe] hydrogenases. The ASA sequence in *Cvu* replaces TSC of all the other Hyd sequences. Ala 223 is aligned to Cys 367 in the [FeFe] hydrogenase of *Clostridium beijerinkii* (CbA5H, PDB code 6TTL [40,41]). In the crystal structure of the latter, Sγ of Cys 367 is found at a small distance from Fe_*d*_ (3.1 Å). In our simulations, the atoms contributing to hinder the access of water to Fe_*d*_ are: the NH group of *adt* ligand (in all simulations); the methyl group of Ala 223 (Cβ, in all simulations); the guanidinium group of Ala 152 (Nη) (in simulations **1** and **2**).

Cβ(Ala 223) and N(*adt*) show similar behavior in simulations **1** and **2**, as they remain stable at distances 4.3 ± 0.3 and 2.5 ± 0.1 Å from Fe_*d*_, respectively. Nη(Arg 152), conversely, is closer to Fe_*d*_ in **2** (3.9 ± 0.8 Å) than in **1** (7.2 ± 1.0 Å), thus explaining the lower accessibility to water of Fe_*d*_ in **2** than in **1** (Figure 5, top-left). Therefore, we find that the Ala 223 and Arg 152 sidechains can act similarly to Cys 367 in the case of CbA5H in limiting water accessibility to Fe_*d*_. As a consequence of this Fe_*d*_ hindering, the reduced state of the H-cluster is partially protected from water soluble O_2_ accessibility. The latter can initiate the H-cluster oxidation, as suggested by previous experimental results [42].

The root-mean square deviation displayed in Appendix A (Appendix A shows that the relaxation of the disordered loops present in all simulations (never included in RMSD calculation) affects more significantly the interactions between the H- and F-domains than the interactions between the structural elements within the H-domain. The simulations (**1** and **3**) were started from the H-domain taken from AlphaFold. They display low RMSD values for both the 11-helix bundle and the two β-sheets, including the case where the reference structure is *Cr*. Simulation **2**, started from an oxidized form of HydA1 in *Cr* (PDB 4R0V [17]), leads to structures different from the available mature crystal structures mainly in the organization of the 7 β strands. In all simulations, the H-domain scaffold does not get distorted in any significant way, as the relative RMSD relative to the initial configurations remains always smaller than 4 Å.

In agreement with the results of the SASA analysis of the different domains and sub-domains (see related discussion in Appendix A), the collected RMSD values indicate that in simulations **1** and **3** structural changes of the initial structures occurs while keeping the protein scaffold almost rigid. Instead N-terminal (F-domain) and loops contribute for the largest part to structural relaxation. However, the whole protein backbone and the H-domain scaffold change more significantly when the F-domain is more structured. This occurs in simulation **3** immediately at the beginning of it and in simulation **2** more slowly during the whole time evolution.

Fluctuations are concentrated on the first few residues of the N-terminus and on the long loop connecting helices h3 and h4 (Arg 168–Cys 224). This loop encompasses a hydrophylic region and the largest fluctuation is around Ala 178, belonging to a 5-Ala segment (Ala 178–Ala 182) in the middle of the loop. Fluctuations are larger in simulation **3** than in **1**, showing a lower stability of the configuration reached after the first 800 ns by **3** compared to that reached in a similar situation by **1**. In particular, the N-terminus is fluctuating even at Cys residues involved in the binding of the F-cluster (Table 1), with a significant stress around Fe atoms. The loop around residue 351 (h7–h8) is also fluctuating. Indeed, the major difference of simulation **2** compared to simulation **1** is in the behaviour of the long h7–h8 loop. In **2**, the reduction of fluctuations in this region increases the fluctuations in the N-terminus. The different behavior of the h3–h4 loop, h7–h8 loop and N-terminus in simulations **1** and **2** appears to be an important marker of mutual interactions between the H- and F-domains. These interactions can be visualized comparing the last configurations obtained after 1 μs as shown in Figure 7.

It can be observed that in simulation **3** the atoms of the [Fe4S4]_*H*_ cluster are left accessible to water (bottom panel), while in the other two simulations water accessibility is hindered by the F-domain. Focusing on the position of the long h3–h4 loop (in green) and C-terminus (in red), one notices that in simulation **1** the F-domain is wrapped around the assembly of the loop and the C-terminus (top-left). This simulation tends to organize N- and C-termini around the long h3–h4 loop. We also notice that the latter is largely embedded into the H-domain, reaching the active site, with Ala 223, that is very close to Fe_*d*_, being the final part of this long loop. In simulation **2**, the N-terminus is more independent by the behavior of the H-domain, and it is less spread over the H-domain and C-terminus (top-right). This condition leaves the N-terminus able to better fold into a nascent β-strand (the yellow ribbons). In simulation **3**, the wrapping of the N-terminal F-domain is diverted, with respect to the case of **1** and **2**, towards other regions of the H-domain. The different disordered segments connecting the structural elements of the H-domain can act as hairpoons towards what, in our case, is the unique external body present in the H-domain environment, i.e., the F-domain.

Summarizing, the comparison between the three simulations indicate that the seven β-strands, organized in short β-sheets, constitute the most rigid portion of the H-domain scaffold. Around this scaffold, at least four disordered regions are arranged: the two terminal regions and two long loops, h3–h4 and h7–h8. We observed that the N-terminus interacts with the H-domain and that, depending on the settling of the H-domain, an F-domain either emerges as β-rich (simulation **2**), tends to release secondary motifs (simulation **1**) or keeps a helical content (simulation **3**). These features indicate inter-domain cross-talk mediated by the disordered regions in the H-domain. From simulation **3**, we can exclude that the F-domain has the structure of *CpI* because this structure provides a non-functional Hyd, with the two clusters at a large distance one from the other. In the other two cases, we observe that a possible binding of the auxiliary F-cluster is consistent with a functional hydrogenase.

### 2.4. Enhancing Statistics

The possibility for the two FeS clusters to be pushed at short distances in simulation **1** indicates the presence of interactions eventually favouring the approach of the F-cluster towards the H-cluster. These interactions seem to emerge when the F-domain is found in a configuration different from that exhibited in the *CpI* hydrogenase.

To sample protein configurations across a broad range of values of the distance between the two FeS clusters, we subjected the system to metadynamics simulations using as an external bias a suitable function of the cluster–cluster distance. By using this method, it is possible to compare the propensity of the two FeS clusters to approach when the F-domain is initially only marginally folded by deep learning method (simulation **1**) to the situation where the F-domain was initially folded as in *CpI* (simulation **3**).

In Figure 8, we compare the free energy as a function of the *d* distance between the two FeS clusters (see Section 4 for details) calculated by using 100 walkers starting with AF (metadynamics **1**) to that calculated by using 100 walkers starting with *CpI*-like configurations (metadynamics **3**). The free energy as a function of the distance between N-terminus and the H-cluster of *Cr* is also shown for comparison.

The comparison between the two *Cvu* curves clearly shows that in the absence of the *CpI* folding of the F-domain ( like in simulation **1**), the minimum of *F* is at about d= 2 nm, while a flat profile around d=2.5 nm is displayed by metadynamics **3**. This comparison confirms that the initial F-domain folding hinders the approach between the two clusters, requiring an extra work of about 50 kcal/mol in order to change the distance from 3 nm to the lowest *d* value of about 0.75 nm (obtained in both metadynamics). Since the short N-terminus of *Cr* allows a larger flexibility compared to *Cvu*, the free energy of *Cr* is flatter. But interestingly a free energy minimum at about *d* = 3 nm is visible and a work only ∼50 kcal/mol lower than metadynamics **1** of *Cvu* is measured as a result of the wrapping of the N-terminus towards the H-cluster in *Cr*.

Some of the structural parameters that have been studied in the equilibrium MD described in the previous sections are discussed below. In Table 3, we display the average values of the scaffold backbone RMSD (using the PDB structures as reference) and of the molecular compactness. Molecular compactness is measured as the ratio *R* between the whole protein SASA and the sum of the SASA of bare F- and H-domains. The two domains were identified in Table 1 and “bare” means that each SASA is computed ignoring the presence of all other atoms. This quantity tends to unit when the two domains do not overlap while it is lower than unit when part of the domains is excluded from the solvent by the presence of the other domain. In the case of hydrogenases, *R* can be slightly larger than unit because of the short independent C-terminus. Average quantities are extracted in different windows of the quantity *d* sampled by metadynamics. The *d*-windows are: 0.75≤d≤1.25 (w1); 1.75≤d≤2.25 (w2); 2.75≤d≤3.25 (w3). We assume that molecules are compact when R≤0.95. The compactness is then measured as the percentage of compact configurations in each *d*-window. We remark that the total number of analyzed configurations is 100,000 for each metadynamics simulation.

It should be noticed that the deviation of the scaffold backbone from the reference structure slightly depends on the value of the chosen collective variable *d*. In *Cvu*, the larger is the distance the lower is the perturbation of the H-domain scaffold. Furthermore, in metadynamics **1** the protein scaffold is less distorted compared to the reference crystal structure than in metadynamics **3**. The *Cr* smaller deviation (RMSD lower than 3 Å) compared to *Cvu* shows that an unstructured N-terminal tail perturbs less the H-domain scaffold than the structured F-domain (metadynamics **3**). The solvent accessibility to the active H-cluster is, on average, in all cases very small, being the maximal SASA of the H-cluster 110 Å^2^ for *Cr*. Most of the *Cr* configurations with H-cluster SASA larger than half the maximum display values of *d* larger than 4 nm. In *Cr*, the H-cluster is known to be accessible to solvent being the enzyme highly sensitive to dioxygen. In *Cvu*, conversely, the H-cluster accessibility is distributed differently. In Figure 9, we display for each configuration the sampled values of *R* and SASA of H-clusters.

The distribution of values shows that in *Cvu* metadynamics **1** compact configurations with low H-cluster accessibility are sampled. In *Cvu* metadynamics **3** these samples appear more rarely, while in *Cr* metadynamics most of the configurations have *R* larger than 0.9 and when R≥0.95 we see, as expected, the highly accessible H-clusters appearing.

In summary, the enhanced statistics metadynamics method shows for *Cvu* an interesting correlation between short distances of the FeS clusters and low accessibility of the active H-cluster to water. These properties are of interest in understanding the minimal architecture of hydrogenase that combines dihydrogen production with resistance to dioxygen. However, the dissection of assemblies between F- and H-domain in case of *Cvu* (or the N-terminal tail and the H-domain in *Cr*) requires long trajectories at constant bias and a proper reweighting of the collected sampling. This analysis will be performed in a following more detailed study.

## 3. Discussion

AlphaFold is now used via almost automated pipelines fed by sequence databases in order to both annotate and propose initial 3D structures. Structural models appear, for instance in UNIPROT, with the prefix AF, usually as proposals of the highest rank. Despite this high-throughput structural prediction available today, a detailed scrutiny of such predictions is necessary.

Therefore, once the function of [FeFe] hydrogenase is associated to the gene KAI3438965.1 of *Cvu*, the information provided by AlphaFold (version 2.2.4) or by other similar softwares needs to be carefully analysed on the basis of detailed chemical properties to be able to truly claim understanding of the possible function of the protein.

The AlphaFold alignment results (see Section 2.1 and Appendix A show that the sequence encoded by gene KAI3438965.1 of *Chlorella vulgaris* has an E-value of zero, 99% query coverage and 74% identity with gene XP_005843810.1 of *Chlorella variabilis* (see Appendix A). The latter *Cv* gene is annotated as region Fehyd_SSU domain-containing, but with unknown function for the coded protein. This unknown protein belongs to the so-called group C of hydrogenases, while we noticed that the known *Cv* Hyd of class A corresponds to gene AEA34989.1. Thus, the above information tells us that at least two [FeFe] hydrogenases can be coded by the *Cv* genome. It is plausible that *Chlorella vulgaris* and *Chlorella variabilis* algae strains contain different [FeFe] hydrogenases like *Clostridium pasteurianum*, that expresses three types of Hyd: *CpI*, *CpII*, and *CpIII* (see Ref. [12] and references therein).

The sequence analysis also shows that the protein encoded in KAI3438965.1 *Cvu* gene is related to proteins that have no hydrogenase function, like Nar proteins of other eukaryotic organisms [33]. However, adding the typical Hyd FeS cofactors to model structures of the protein expressed by *Cvu* KAI3438965.1 gene, we showed that one of the resulting constructs is consistent with a functional hydrogenase. It is therefore possible that the same protein turns into a hydrogenase when internalized into the chloroplast where the H-cluster is completed with the [2Fe]_*H*_ component.

Different *Chlorella vulgaris* strains have been proposed as interesting candidates for sustainable biological hydrogen production [6,25,43]. Despite the phenotypical characterization available for these microalgae, only the genome of the 211/11P strain is published [5]. Moreover, the structural information about [FeFe] hydrogenases of microalgae is poor. In particular, the *Chlamydomonas rheinardtii* crystal and solution structures available in the Protein databank do not contain complete FeS clusters [14,44]. In any case, *Cr* Hyd lacks a functional F-domain and this missing domain can be responsible for the low resistance of *Cr* Hyd to oxidation [9].

*Chlorella vulgaris* 211/11P [FeFe] hydrogenase could be a very important system for experimental characterization of the type of cooperation between the single [4Fe4S] auxiliary cluster and the H-cluster [9]. The interactions of the iron–sulfur catalytic site (H-cluster) with the local protein environment are thought to contribute to modulate catalytic reactivity, but this has not been fully demonstrated. Some studies indicate that protein secondary interactions directly influence the relative stabilization/destabilization of different oxidation states of the metal cluster active site [2].

In some microalgae organisms, the evolutionary process has guided [FeFe]-hydrogenase to lose the F-domain and replace it with a smaller disordered domain of variable sequence [11]. Interactions between the F-domain and the H-domain modulate the accessibility to the H-cluster of O_2_ and other reactive oxygen species, irrespective of their generation mechanism, thus eventually decreasing the rate of protein oxidation [42,45]. The possible protective effect depends on the mutual interactions between F- and H-domains which must favour a short distance between F- and H-clusters if it has to allow electron transfer from the exogenous reductant to Fe_*d*_ in the H-cluster. In this work, we investigated the possible protective effect of the Hyd N-terminus which is able to mimic in short microalgae Hyd the large F-domain that is present in cyanobacteria.

To investigate this possibility, we built three models of pseudo F-domains in *Cvu*. We showed that a structure of the F-domain similar to *CpI* provides a non-functional hydrogenase, because the two FeS clusters are too far apart in space to transport electrons from reductant species to the active H-cluster. Conversely, alternative structures produced by a deep learning algorithm provided a pseudo F-domain that can be functional and at the same time can hinder the H-cluster to O_2_ accessibility. In another model proposal the additional FeS cluster in the F-domain is also accessible to external reductant species.

We have shown that the combination of sequence alignment, deep learning algorithms, and atomistic molecular dynamics simulations with the inclusion of inorganic cofactors, has the potential of becoming a new pipeline for model studies addressed at understanding and lately optimizing the function of [FeFe] hydrogenase.

## 4. Materials and Methods


For brevity, the acronyms listed at the end of the text are used in next section.

### 4.1. Alignment and Gene Annotation

The *Cvu* genome has been published with the code SIDB01000001.1 in Genbank [5]. To identify the protein with a candidate hydrogenase function, we aligned the whole genome of *Cvu* to the known sequences of 7 [FeFe] hydrogenase: *Clostridium pasteurianum*, *Desulfovibrio desulfuricans*, *Chlamydomonas rheinardtii*, *Chlorella variabilis*, *Chlorella* sp. DT, *Chlorella fusca*, *Tetradesmus obliquus*. We performed BLAST alignments [46,47] using BLOSUM90 [48] and PAM30 matrix [49,50]. Then, we singled out and extracted the sequence KAI3438965.1 that always shows the highest score in all alignments. We compare and summarize the results obtained with BLAST [51] and CLUSTAL [52] algorithms and are reported in Appendix A.

### 4.2. Structure Prediction

In order to generate a hypothetical *Cvu* Hyd with one auxiliary [4Fe4S] cluster in the F-domain, we proceeded as explained below.

In the first case (**1**), in all the AlphaFold [20,53] (version 2.2.4) predicted structures we looked for a configuration where Cys residues 21, 72, 75, and 78 were able to bind the F-cluster (see sequence alignment in Figure 4). Then we inserted the auxiliary [4Fe4S]_*F*_ cluster and minimized the energy once a suitable force-field for the system is identified (see Structure refinement below). The second model (**2**) comes from SwissModel, a fully automated protein structure homology-modeling server [19]. We choose SwissModel prediction “two”, because prediction “one” was obtained from AF which is integrated into SwissModel, and it is, therefore, identical to **1**. In **2**, only a partial H-domain from residue 86 to 540 is present. Therefore, we completed the structure with the prediction made by AF for the missing residues. Once the H-domain scaffold is aligned and overlapped (see Appendix A), we obtain a full model structure where residues 1–100 and 531–549 are taken from the AF prediction and residues 101–530 from the SwissModel prediction.

In the third case (model **3**), we substituted the unfolded F-domain of the AF prediction having the highest plDDT score, with the folded F-domain taken from 6N59 PDB structure of *CpI* [12]. To do so, we first replaced the coordinates of atoms in residues 1–88 of *Cvu* with the coordinates of those atoms that are shared with residues 119–206 in 6N59. We then minimized the relative root-mean square deviation (see below) of the atoms in residue 82 with respect to the same residue in *Cvu* as generated by AF. In this way, we obtain a model in which the H-domain is taken from AF and the F-domain is mapped to the coordinates of the crystal structure of *CpI*. The initial structures of the three models are displayed in Appendix A.

As a result of this procedure, we obtained three model structures with very similar H-domain scaffolds, but a variety of F-domains and disordered loops.

### 4.3. Structure Refinement

After the initial structure prediction (see previous sub-section), in all models we bonded all the clusters inside the protein matrix as in the H-ox state: the [4Fe4S]_*H*_ cluster has the charge 2+, corresponding to the formal 2Fe(II)2Fe(III) oxidation state of iron ions; the [2Fe]_*H*_ cluster has the charge −1, corresponding to the formal Fe(I)Fe(II) oxidation states of the two iron ions; the auxiliary [4Fe4S]_*F*_ cluster has the same parameters of [4Fe4S]_*H*_.

In order to insert the clusters and run MD simulation, we used the available force-field parameters of FeS clusters [54] that we adapted to the Charmm36 force-field for proteins [55] (see Appendix A, the Force_Field folder). The FeS clusters are explicitly bonded to the protein matrix, with the bonds between Cys residues and Fe atoms described by the chosen force-field (see below). We inserted the protein (7957 atoms) in a simulation cell, adopting periodic boundary conditions, with water plus ions atoms to neutralize the system ([NaCl] = 0.15 M). All the three systems are made by ∼10^5^ atoms. After an initial energy minimization and equilibration in NVE statistical ensemble (20 ps), the systems were equilibrated for 100 ps in the NVT ensemble first at a temperature of T=150 and than at T=300 K. The configurations of the three models obtained after energy minimization are displayed with all the bonded FeS clusters in Appendix A (see Appendix A).

The equilibration was continued for another 2 ns in the NPT ensemble, at a constant T=300 K with a bath relaxation time of 0.1 ps. The pressure was kept at 1 bar with a barostat relaxation time of 2 ps and compressibility of bulk water at room conditions.

Coulomb and Lennard-Jones interactions were described by means of a buffered Verlet pair list with a short-range cutoff equal to 1 nm [56]. Long-range electrostatic interactions were treated with the particle mesh Ewald method [57] with a grid spacing of 0.16 nm. The LINCS [58] and SETTLE [59] algorithms were employed to constrain, respectively, all protein bonds involving H atoms and internal degrees of freedom of water molecules, allowing to integrate the equations of motion with a 2 fs time-step. Finally, starting from the equilibrated systems, we performed 1 μs long MD simulations for each of the three different models in the NPT ensemble, initiated with different random seeds to generate the initial atomic velocities from a Maxwell distribution at 300 K. The simulation orthogonal cell’s sides ($L_{x}$, $L_{y}$, and $L_{z}$) were, on average after 800 ns of equilibration for simulation **1**: $\langle L_{x}\rangle =11.330\pm 0.006$ nm; $\langle L_{y}\rangle =7.352\pm 0.004$ nm; $\langle L_{z}\rangle =12.496\pm 0.006$ nm, for simulation **3**: $\langle L_{x}\rangle =10.858\pm 0.006$ nm; $\langle L_{y}\rangle =7.9678\pm 0.004$ nm; $\langle L_{z}\rangle =10.437\pm 0.006$ nm.

### 4.4. Enhancing Statistics with Well-Tempered Metadynamics

Many computational methods have been devised to address the well-known limitations of equilibrium molecular dynamics, that is the insufficient sampling of configurations of macromolecules when keeping the conditions of thermal equilibrium. Among the many techniques used to enhance sampling, we decided to use altruistic multiple-walkers well-tempered metadynamics [60,61]. The method defines a collective variable (CV), that we denoted as ξ, function of atomic positions, *q*. The values, *s*, taken by ξ can be used to label the system macrostates. The set of coordinates *q* labels the system microstates, with each set of *q* yielding one of the possible values of *s*. If ergodicity holds, infinitely long simulations of a trajectory q(t) in a given statistical ensemble would correctly sample the statistical weight of *s*. However, because of the huge number of ways in which certain values of *s* of ξ are encountered, compared to others, actual numerical simulations in practice only sample the maximally degenerate values of ξ. This is the case where ξ is the CV associated to folding/unfolding events. Generalized statistical ensembles try to address this problem by biasing the trajectory to spend more time where ξ has a low degeneracy and less time where ξ has a large degeneracy. The sampling of configurations obtained with the bias of the inverse probability of ξ is called metastatistics.

In our application, the enhanced statistics (metastatistics, hereafter) is obtained from a swarm of trajectories (called walkers) provided by metadynamics while building up a suitable external bias, which is then kept fixed when collecting configurations in the final step of the NpT simulation. The time evolution of each of the walkers starts from an initial configurations built either on the basis of AF predictions, (metadynamics **1**) or manual construction (metadynamics **3**). The initial configurations of each walker are different mainly because of the different interactions with the macromolecular environment. The initial differences are then amplified by the different construction of the external bias so that a variety of configurations among the many possible ones labelled by the values of the collective variable ξ are sampled.

The external bias that induces ξ broadening is progressively built adding repulsive gaussian functions of the collective variable term in the potential. In well-tempered metadynamics [62], as the simulation proceeds, the width of the added Gaussian remains constant but its height steadly decreases. The bias magnitude, which increases monotonically, eventually changes very little with time. As we said, at the beginning the space of CV is flooded by gaussians of height *w*. With the progress of flooding, the height of the newly added Gaussians decrease. This behavior is very important in highly complex biological systems, where the biasing potential should never reach any excessively large value.

In this work, we used a biasing factor $\frac{T+\Delta T}{\Delta T}=$16 (ΔT= 20 K) for *Cvu*, and 8 for *Cr* molecules. In order to sample both active and inactive structures for *Cvu*, we choose as CV the distance between the center of mass of the two [4Fe4S] clusters, one in the H-cluster and the other in the F-cluster. In particular, there is an effective catalytic function when the CV is inside a specific range (11–14 Å).

In the case of *Cr*, where the F-cluster is not present, we take as CV the distance between the center of mass of the single [4Fe4S] cluster and the N-terminus of the molecule.

While only the highest-score AlphaFold prediction was used in conventional simulation **1**, in metadynamics **1** we used the five highest-score AlphaFold predictions. Each of the 5 configurations was randomly rotated 19 times and then inserted into an orthorhombic simulation cell. We thus obtained 5 × 20 configurations, each used as starting configuration of a walker. In metadynamics **3**, we used the single manual construction randomly rotated 100 times. The bias construction, providing the free energy as a function of the chosen collective variable *d*, lasted 28 ns. Gaussian functions were added to the bias every 10 ps. The bias exchange among the 100 walkers was performed every 2 ns. The free energy displayed in Figure 8 is the opposite of the converged bias. Data reported in Table 3 are obtained averaging over the last 10 ns simulation collected at constant bias after convergence and over all 100 walkers. Averages are not reweighted to cancel the bias because at this stage we want to describe the behavior of configurations sampled at different *d* values.

### 4.5. Analysis

In order to compute average quantities in conventional MD **1**–**3**, we used the last 200 ns of each simulations. Structural quantities of interest are described below. RMSD measures the average distance between structure at time $t_{2}$ with respect to a reference structure at time $t_{1}$. The definition of RMSD is:(1)$$\mathrm{RMSD}\left(t_{2},t_{1}\right)=\sqrt{\frac{1}{M}\sum_{i=1}^{N_{atoms}}m_{i}{\left|r_{i}\left(t_{1}\right)-r_{i}\left(t_{2}\right)\right|}^{2}}\;,$$
where *M* is the total mass of the molecule, $m_{i}$ and $r_{i}$ are the mass and position of atom *i*. The structure at time $t_{2}$ is standardly translated and rotated with respect to the reference one so as to minimize the RMSD. RMSD is useful to compare two structures and look for the differences along the simulation.

Root-mean square fluctuation (RMSF) instead measures the average distance between atoms belonging to the same residue *i*, averaged over time. In our analysis we computed such quantity as emerging from a sub-set of eigenvectors of the covariance matrix, as it is routinely made in principal component analysis (PCA) [63]. To simplify the analysis we used the first 4 eigenvectors (descending order of eigenvalues) of the covariance matrix. Also, we used the backbone heavy atoms to compute the covariance matrix, to focus the study on major structural changes of the protein

The gyration radius ($R_{g}$) is defined by the formula
(2)$$R_{g}\left(t\right)=\sqrt{\frac{1}{M}\sum_{i=1}^{N_{atoms}}m_{i}{\left|r_{i}\left(t\right)-r_{CM}\right|}^{2}}\;,$$
where CM identifies the center of mass of the molecule.

In order to characterize the accessibility of the solvent to protein surface and clusters atoms, we calculated the solvent accessible surface areas (SASA) [64] and the radial distribution function (g(r)) of water molecules, respectively. The SASA is measured with a spherical probe mimicking a water molecule (with a radius of 1.4 Å). Standard atomic radii [65] are used for protein atoms. In SASA calculations Fe is assumed to have zero radius since it is buried by ligand atoms. The function g(r) is defined as:(3)$$g\left(r\right)=\frac{1}{{\langle \rho_{b}\rangle }_{local}}\frac{1}{N_{a}}\sum_{i\in a}^{N_{a}}\sum_{j\in b}^{N_{b}}\frac{\delta (r_{ij}-r)}{4\pi r^{2}}\;,$$
where δ is the Dirac function and ${\langle \rho_{b}\rangle }_{local}$ the density of atoms of type *b* averaged over all spheres of radius *r* around atoms of type *a*.

Finally, salt bridges are the number of contacts between two groups of atoms, *a* and *b* with opposite charge. Choosing *a* as Cδ(Glu), Cγ(Asp), C-terminus and *b* as Nζ(Lys), Nη(Arg) and N-terminus, we define
(4)$$\mathrm{SB}=\sum_{i\in a}\sum_{j\in b}S_{ij}\;\;$$
where $S_{ij}$ is the step function, which is equal to 1 if the contact (distance ≤ 4 Å) between atoms *i* and *j* is formed and zero otherwise.

All MD simulations were performed with the NAMD 2.13-14 [66] and GROMACS 2021.5 [67,68] codes. Analysis was performed with GROMACS 2021.5, VMD 1.9.3 [69] and PLUMED 2.8.1 [70] softwares. The TopoGromacs Tool [71] was used to convert Charmm topology and parameters into Gromacs input data-sets.

## 5. Conclusions

In this work, we have presented a computational study of the *Chlorella vulgaris* 211/11P putative [FeFe] hydrogenase. The unicellular algal strain produces, under suitable conditions, pure hydrogen gas by using sunlight as a primary energy source and water as a hydrogen source. The system we are studying is challenging even for advanced annotation and structure prediction methods, because of the need for also including non-protein cofactors correctly bound to the protein.

We found that sequence alignment analysis, AlphaFold prediction models, and simulations including the required metal clusters, are consistent with the data published in the literature [2,9,12,13,14,37]. Our analysis led us to conjecture that the *C. vulgaris* 211/11P strain [FeFe] hydrogenase belongs to a novel family of [FeFe] hydrogenases. The main points on which we base our conclusion that the above represents a new hydrogenase are: (i) it consists of a functional H-domain, with the peculiar H-cluster binding motif ASACPGW replacing the conserved TSCCPGW motif of most of all the other known [FeFe] hydrogenases; (ii) it can contain a single auxiliary 4Fe4S cluster bound to a non-conserved F-domain; (iii) depending on the F-domain degree of structural arrangement, significant effects on the structural parameters are observed; (iv) we can exclude that the potential F-domain has the structure of [FeFe] hydrogenase I in *Chlostridium pasteurianum*.

Taken together, the models we investigated reveal important properties of the [FeFe] hydrogenase of *Chlorella vulgaris* 211/11P strain, highlighting the need for further investigation of the interactions between F- and H-domains. Indeed, understanding the cross-talk between the two domains and, in general, between disordered and ordered regions, is necessary to optimize the hydrogen production efficiency of these enzymes.

## Figures and Tables

**Figure 1 ijms-25-03663-f001:**
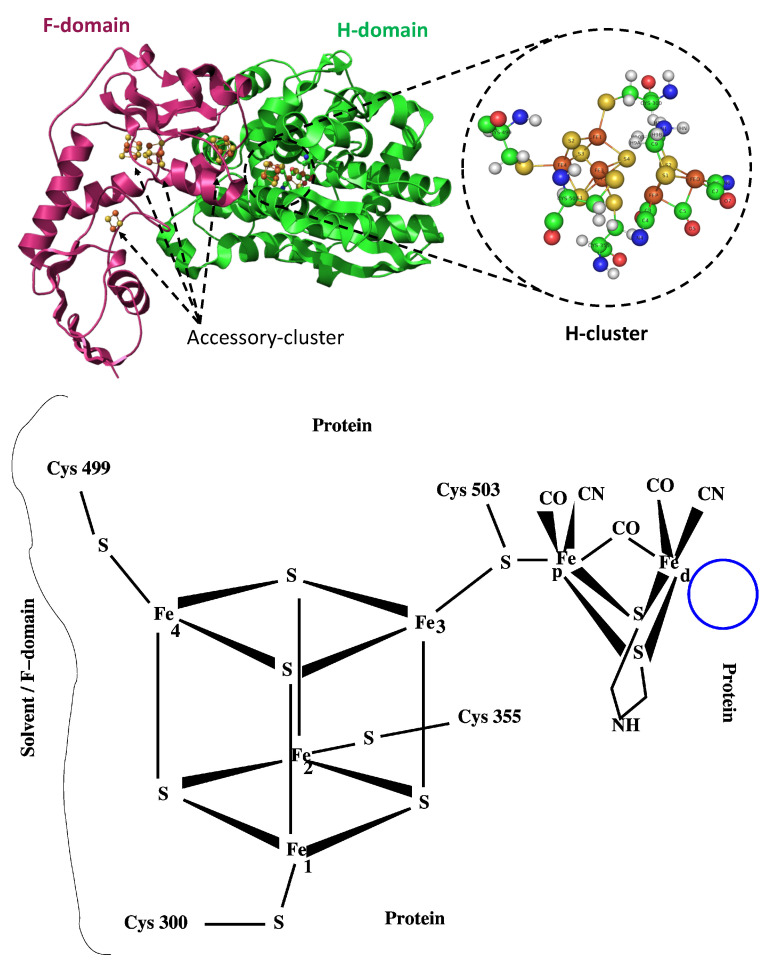
(**Top panel**)—[FeFe] hydrogenase crystal structure of *CpI* in bacteria *Clostridium pasteurianum* (PDB code: 6N59): H-domain and F-domain are represented as green and pink ribbons, respectively, with PyMOL [10]. H-cluster and auxiliary clusters are explicitly indicated. (**Bottom panel**)—A sketch of the chemical structure of the H-cluster. The blue circle identifies the region close to Fe_*d*_ accessible to water molecules through channels in the protein matrix.

**Figure 2 ijms-25-03663-f002:**
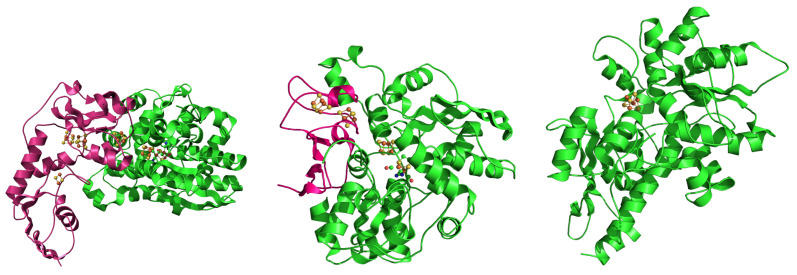
Three structures determined by X-ray crystallography of [FeFe] hydrogenase. (**Left**): *Cp* (HydA1 gene, *Clostridium pasteurianum* (bacteria), PDB 6N59 [12]). (**Middle**): *Dd* (HydAB gene, *Desulfovibrio desulfuricans* (bacteria), PDB 1HFE [13]). (**Right**): *Cr* (HydA1 gene, *Chlamydomonas rheinardtii* (algae), PDB 3LX4 [14]). Secondary motifs assigned with STRIDE [15] are displayed as cartoons with the use of the PyMOL program [10]. FeS clusters are displayed as spheres (S is yellow, Fe is orange). The proteins are not in scale. The size of the proteins decreases from left to right. The images are rescaled to fit into each frame. The F-domain is emphasized in pink color.

**Figure 3 ijms-25-03663-f003:**
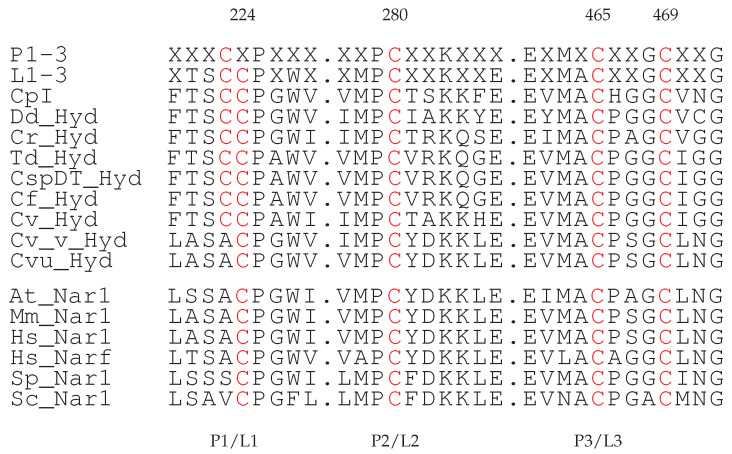
Hyd fingerprint motifs (P1–3 of Ref. [16], L1–3 of Ref. [29]) in 8 different species (all species are in Appendix A). The last 6 rows are the regions aligned to P1–3 and L1–3 fingerprints in Nar (see main text). Aligned Cys residues are those with numbers (top) as in the *Cvu* sequence. Abbreviations for organisms (see also Section Abbreviations) are Cf, *Chlorella fusca*; CspDT, *Chlorella* species, DT strain (Taiwan); Td, *Tetradesmus obliquus*; At, *Arabidopsis thaliana*; Sc, *Saccharomyces cerevisiae*; Sp, *Schizosaccharomyces pombe*; Hs, *Homo sapiens*; Mm, *Mus musculus*. *Cv_Hyd* and *Cv_v_Hyd* are the proteins expressed by gene AEA34989.1 and XP_005843810, respectively, in *Cv*.

**Figure 4 ijms-25-03663-f004:**
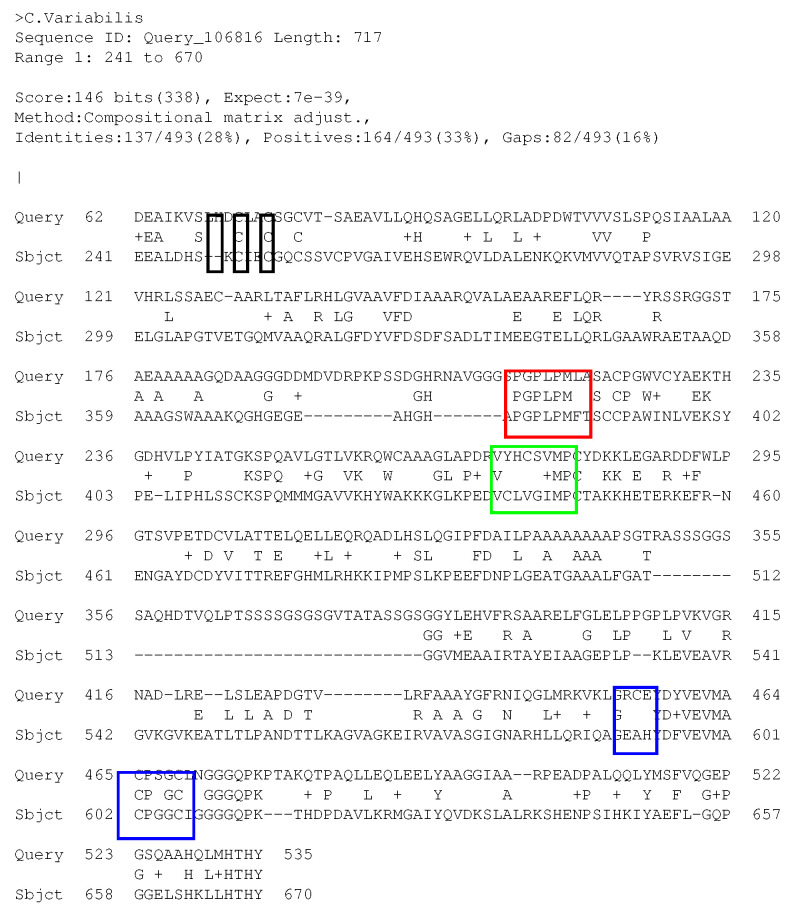
Sequence alignment between *Cv* Hyd (Sbjct) and *Cvu* Hyd (Query). Motifs P1, P2, and P3 (see Figure 3) are enclosed within red, green, and blue boxes, respectively. The Cys-rich motif in the N-terminus is enclosed within black boxes.

**Figure 5 ijms-25-03663-f005:**
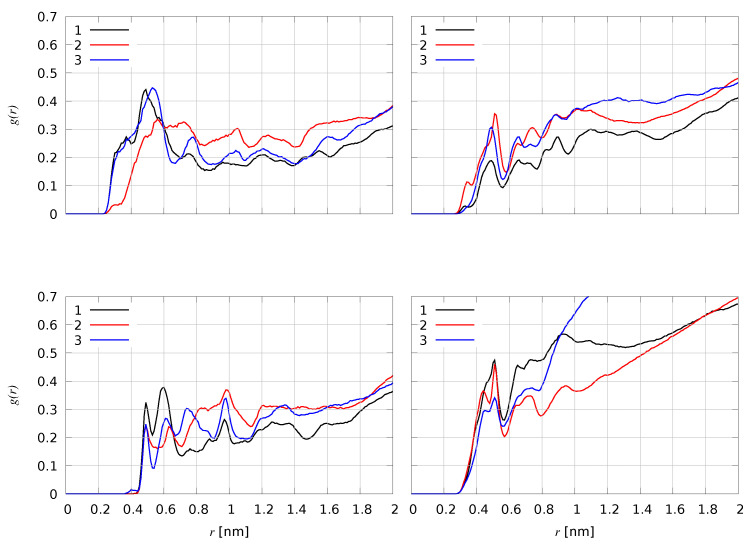
Radial distribution function of Fe-Ow pairs, with Ow oxygen atoms belonging to water molecules and Fe belonging to different FeS clusters. Black curves: simulation **1**. Red curves: simulation **2**. Blue curves: simulation **3**. (**Top-left**): Fe is Fe_*d*_ (distal Fe atom in [2Fe]_*H*_). (**Bottom-left**): Fe is Fe_*p*_ (proximal Fe atom in [2Fe]_*H*_). (**Top-right**): all Fe atoms belonging to [4Fe4S]_*H*_. (**Bottom-right**): all Fe atoms belonging to [4Fe4S]_*F*_. Averages are computed over the last 200 ns.

**Figure 6 ijms-25-03663-f006:**
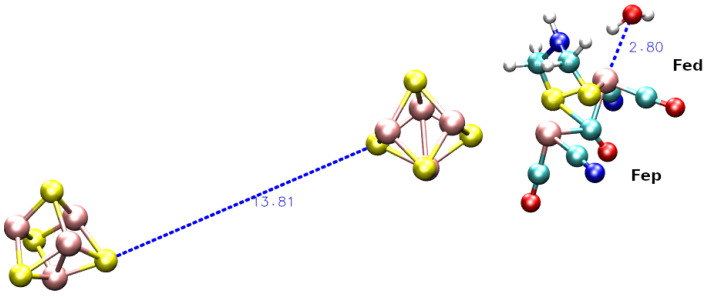
F-cluster and H-cluster after 800 ns in simulation **1**. FeS clusters are displayed as spheres, with Fe in pink and S in yellow. C is cyano, N blue, O red, H white. The atomic dimension is given by the van der Waals radius of each atom. Blue dashed lines display the distance vectors with the moduli as numbers in Å.

**Figure 7 ijms-25-03663-f007:**
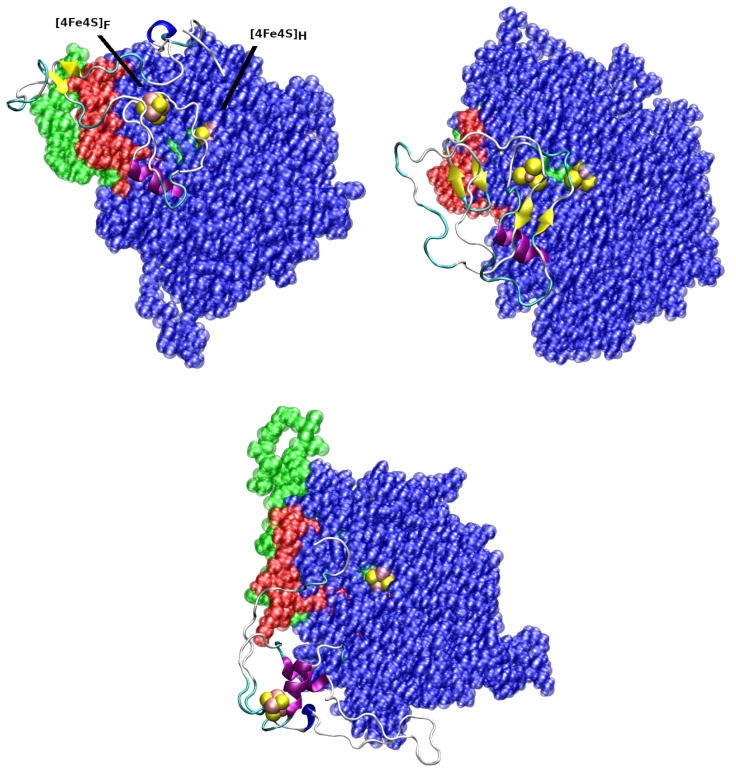
*Cvu* Hyd after 1 μs of MD simulation. (**Top-left**): simulation **1**. (**Top-right**): simulation **2**. (**Bottom**): simulation **3**. The H-domain is displayed as a set of blue transparent spheres. The disordered h3-h4 loop (residues Arg 168–Cys 224) is in green. The C-terminus (residues Met 531–Trp 549) is in red. The F-domain is shown as a cartoon representing the secondary motifs calculated by STRIDE. FeS clusters are displayed as spheres, with Fe in pink and S in yellow. The atomic dimension is given by the van der Waals radius of each atom. Residues His 89-Leu 530 (H-domain) have been translated and rotated in order to minimize the magnitude of the relative RMSD of heavy atom backbone of the three structures.

**Figure 8 ijms-25-03663-f008:**
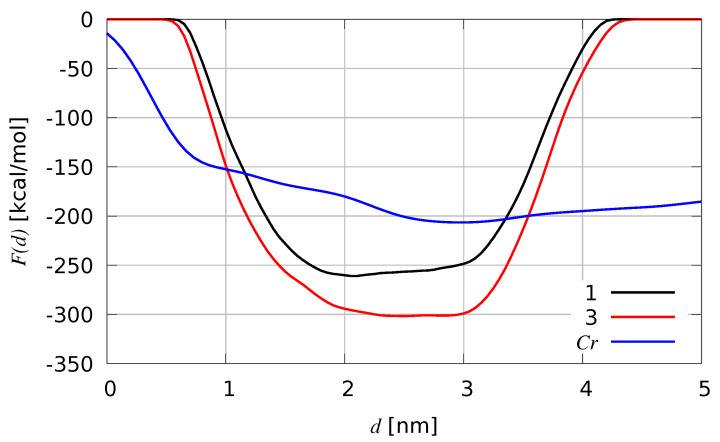
Free energy (*F*) as a function of the distance between FeS clusters’ centers of mass (*d*) for *Cvu* Hyd, after bias build-up convergence in multiple-walkers metadynamics. The same function of the distance between N-terminus and H-cluster in *Cr* is shown for comparison. Black curve: metadynamics **1**. Red curve: metadynamics **3**; Blue curve: metadynamics of *Cr*.

**Figure 9 ijms-25-03663-f009:**
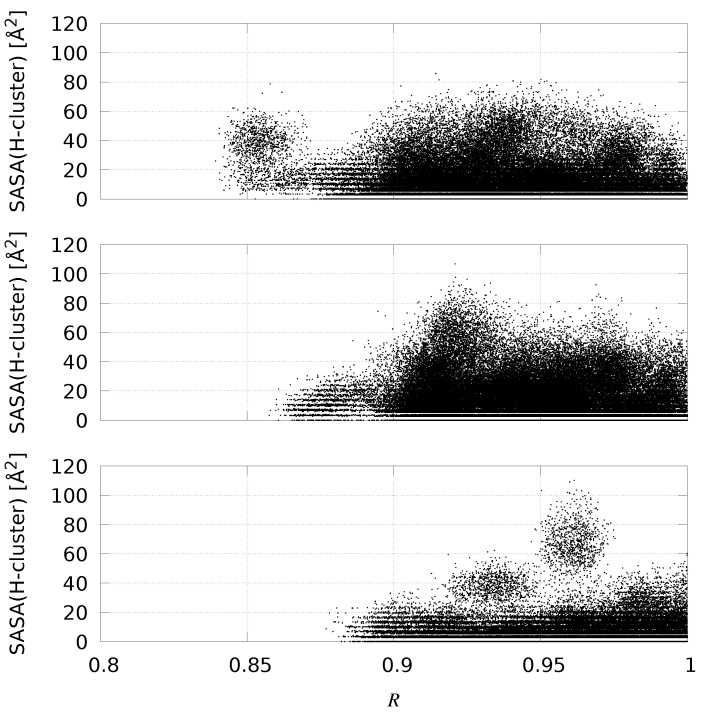
Sampled points in *R*-SASA(H-cluster) space for *Cvu* and *Cr* Hyd. Top–*Cvu* metadynamics **1**. Middle–*Cvu* metadynamics **3**. Bottom–*Cr* metadynamics.

**Table 1 ijms-25-03663-t001:** Summary of protein domains conserved among Hyd of different species. The numbers represent the location of the Cys residues that bind FeS clusters. In the last line, we show our hypothesis for the location of the *Cvu* F- and H-domain and C-terminus.

Species	F-domain	H-domain	C-terminus
	**Aux. clusters**	**H-cluster**	
*CpI*	1–209	210–568	569–574
	157, 190, 193, 196	300, 355, 499, 503	
	147, 150, 153, 200		
	94, 98, 101, 107		
	36, 46, 49, 64		
*Dd* Hyd	1–86	87–392	393–421
	45, 66, 69, 72	178, 234, 378, 372	
	35, 38, 41, 76		
*Cr* Hyd	1–64	65–487	488–497
		170, 225, 417, 421	
*Cvu* Hyd	1–88	89–530	531–549
	21, 72, 75, 78	224, 280, 465, 469	

**Table 2 ijms-25-03663-t002:** RMSD (Å) of heavy backbone atoms (N, Cα, C, O), belonging to different sets of aligned residues. The left value is for the whole H-domain scaffold, middle is for the 11 helix-bundle, right is for the 7 β-strands. *Tm* is Thermotoga maritima. **1** indicates here the initial configuration generated by AlphaFold.

Reference	*CpI*	*Dd*	*Cr*	*Tm*	*Cvu*
**Target**		**Hyd**	**Hyd**	**HydABC**	**Hyd** **1**
*CpI*	0				
*Dd* Hyd	1.7/1.8/0.8	0			
*Cr* Hyd	3.2/3.6/0.7	3.0/3.4/0.9	0		
*Tm* HydABC	2.2/2.3/1.6	2.3/2.4/1.6	3.7/4.2/1.5	0	
*Cvu* Hyd **1**	22.2/4.5/2.0	22.2/4.2/2.2	22.8/5.4/1.7	22.4/4.8/2.5	0

**Table 3 ijms-25-03663-t003:** RMSD and compactness measured for different sets of configurations. Reference structure are 6N59 (chain A backbone) and 3LX4 (chain A backbone) for *Cvu* and *Cr*, respectively. Compact samples are the percentage of those in the given window with R≤0.95 over the whole number of collected configurations (100,000).

Configurations	〈RMSD〉 (Å)	Samples in Set	〈SASA(H-Cluster)〉	Compact Samples (%)
**1**/w1	3.6	31,217	1.8 ± 1.6	18
**1**/w2	3.5	6733	1.7 ± 1.1	4
**1**/w3	3.4	4220	1.3 ± 1.4	1
**3**/w1	4.0	43,305	1.6 ± 1.5	31
**3**/w2	3.9	4410	1.0 ± 0.7	2
**3**/w3	3.7	240	1.9 ± 0.7	0
*Cr*/w1	-	0	-	-
*Cr*/w2	2.7	14	1.7 ± 1.0	0
*Cr*/w3	2.8	1751	0.9 ± 0.9	1

## Data Availability

Part of the data to reproduce the reported calculations are available online at: https://doi.org/10.3390/ijms25073663. A document, SM.pdf, in Appendix A explains how to use the provided files deposited in Appendix A. Any further data useful to reproduce the reported calculations are available on request to the corresponding author.

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
