# Peer review of "Predicting the Structure of Enzymes with Metal Cofactors: The Example of [FeFe] Hydrogenases"

_ijms, 2024, doi:10.3390/ijms25073663_

Round 1
Reviewer 1 Report (Previous Reviewer 1)
Comments and Suggestions for Authors
The article “Predicting the structure of enzymes with metal cofactors: The example of [FeFe] hydrogenases” is devoted to the study of the structure of the enzyme [FeFe] hydrogenase from the microalgae Chlorella vulgaris 211/11P, which plays an important role in the production of molecular hydrogen. This enzyme has biotechnological potential. In the article, using modern molecular modeling methods, model structures of this protein were obtained. The dynamics of these models was studied using classical molecular dynamics, principal component analysis and metadynamics. This paper is recommended for publication after minor revision.
The main question for the article is this. Structural models derived from deep learning simulations do not always predict experimental structures well. (See, for example, https://www.mdpi.com/2218-273X/13/11/1611 (Figure S1), https://www.mdpi.com/1424-8247/16/12/1662) Have there been any attempts to model the proteins presented in the article by homology?
Unfortunately, the actual reliability of the model can most likely only be determined by the confirmed or unconfirmed three-dimensional structure of the protein, resolved experimentally. Until the structure is obtained, the quality of the model as a natural protein remains open. However, thanks to the use of modern modeling methods and from a methodological point of view (including extensive supplementary material), the article is certainly interesting and provides important information about the interaction of the cofactor with the protein matrix of [FeFe] hydrogenases.
Line 11. Chorella vulgaris 211/11P -> Chlorella, add “l” letter
Figure 2. Three structures determined by X-ray crystallography of [FeFe] hydrogenase.
Add the names of living organisms that correspond to the proteins shown in the figure.
Line 311 on the loop around residue 178, connecting helices h3 and h4. Fluctuations are larger in 311
Please provide the names of amino acid residues corresponding to the numbers. For example, Ala178, etc.
Figures 4, 6, 9
Please make the lines thicker, otherwise they will get lost behind the grid lines.
4.3. Structure refinement
How were the FeS clusters held in the protein matrix during the simulations?
Author Response
We answered all points raised by reviewer, see marks A1.x in the revised manuscript.
See also page 2 of attached cover letter.

Reviewer 2 Report (Previous Reviewer 3)
Comments and Suggestions for Authors
The work presented for review, only cosmetically differs from the original version of the manuscript. The corrections made by the authors do not remove the basic flaws of the material presented: a) the presentation of information known to the scientific world as original, when such information is already in scientific circulation (annotation of the amino acid sequence); b) the study of a theoretical structural model, built without due diligence and on the basis of all possible sources of information.
# A)
The authors mention in the modified manuscript (Lines 122-128) that the annotation of the studied protein in the UNIPROT database appeared in June 2023. One can easily trace the entire modification history of the E1ZR28 record (www.uniprot.org/uniprotkb/E1ZR28/history)
In the UNIPROT database, the record E1ZR28 (studied protein) appeared on November 30, 2010
In the first modification of the record (January 11, 2011) appears the first annotation qualifying the sequence as belonging to the Fe-hydrogenase family
DR EMBL; GL433861; EFN51708.1; -; Genomic_DNA.
DR InterPro; IPR009016; Fe_hydrogenase.
DR InterPro; IPR004108; Fe_hydrogenase_lsu_C.
DR InterPro; IPR003149; Fe_hydrogenase_ssu-like.
DR Gene3D; G3DSA:4.10.260.20; Fe_hyd_ssu-like; 1.
DR Pfam; PF02906; Fe_hyd_lg_C; 1.
DR Pfam; PF02256; Fe_hyd_SSU; 1.
DR SMART; SM00902; Fe_hyd_SSU; 1.
DR SUPFAM; SSF53920; Fe_hydrog; 1.
rest.uniprot.org/unisave/E1ZR28?format=txt&versions=2
As of the 26th modification (June 31, 2019, the name of the protein is "Fe_hyd_SSU domain-containing protein". rest.uniprot.org/unisave/E1ZR28?format=txt&versions=26 As of the 34th modification of the record (February 22, 2023), the name of the protein is "Full=Iron hydrogenase small subunit domain-containing protein". rest.uniprot.org/unisave/E1ZR28?format=txt&versions=34 I don't know when the authors started working on the manuscript, but for almost a year the protein has had the full annotation "Full=Iron hydrogenase small subunit domain-containing protein." Please note that the earlier name of the protein used since June 2019 is an acronym for the current name, and this is only due to the naming system used by the UNIPROT database. In biology, it is very common to use multiple names to denote paralogous sequences found in different organisms. The role of the scientist is to collect and select data. In the NCBI database, the record KAI3438965.1 is further described as "hypothetical protein D9Q98_001379" does not mean that nothing is known about this protein. The KAI3438965.1 record in the NCBI database was created on May 12, 2022 and has not been modified since then. Despite this, the contents of the record contain information about the so-called regions. Two regions "Iron only hydrogenase large subunit, C-terminal" and ""Iron hydrogenase small subunit; smart00902" are described. These regions unambiguously classify the sequence as a hydrogenase. At the same time, these regions correspond to what the authors in the manuscript show in Table 1. In the Conserved Domains Database (www.ncbi.nlm.nih.gov/Structure/cdd), one can see that these conserved regions are represented in a very wide range of amino acid sequences (www.ncbi.nlm.nih.gov/Structure/cdd/cddsrv.cgi?uid=397172, www.ncbi.nlm.nih.gov/Structure/cdd/cddsrv.cgi?uid=214899). Perhaps this information will help build a more likely structural model of the protein under study. # B) The protein structural model presented and tested is highly problematic. The model derived from prediction using the AlphaFold algorithm contains very long regions of sequences that do not have a defined spatial structure (Model 1). The authors also present Model 2 which is a hybrid, part of the structure (H domain) comes from modeling with AlphaFold software, while the N-terminal part of the protein (F domain) was built by the authors. For a moment, I will focus on modeling domain H. Using a very simple tool like SWISS-MODEL (swissmodel.expasy.org), one can get very nice compact and very reliable structures of domain H. The model is based on templates (PDB ID 8A6T, 4ROV, 8QM3). The structures of the models are compact, free of regions of undefined structure. I suggest the authors to use other methods for protein structure prediction than just AlphaFold. There are many such methods and tools. It seems that modeling the structure of the F domain is very challenging, due to the complete lack of homologous sequences. The method used by the authors to model the structure of the F domain (model 2) completely ignores amino acid sequence homologies. In addition, the authors used a single amino acid residue as the connection point between the two domains, which may result in their completely random orientation. This random orientation may result in excessive distances between the FeS clusters observed in the simulation (see Figure 7). Perhaps modeling the H-domain separately (see text above) and using the F-domain model presented in the paper, and attempting to reorient the two domains spatially relative to each other would yield some model that corresponds to known experimental facts (distances between FeS clusters). I hope that the information presented above will allow the authors to develop a new version of the manuscript that will contain, original, reliable scientific data of interest to the scientific community.

None
Author Response
We made our best to answer the criticism of reviewer.
We included a new model, as suggested by reviewer, in the revised manuscript.
Results section was extensively changed.
See also page 3 of attached cover letter.

Round 2
Reviewer 2 Report (Previous Reviewer 3)
Comments and Suggestions for Authors
Supplementary data as well as references to them in the text should be sorted out before publishing the paper. "By inspecting the global alignments (see SM) we observe that also the Cys residues in" line 165. It is not entirely clear what the authors are referring to, the references must be precise. Here, they are probably referring to the Blast_Clustal_Alignments directories (this is only a presumption of the reviewer), but this directory contains many files and it is not clear what the authors are referring to. Some of the additional data is in one well-described SM.pdf file and in directories with defined contents. However, in addition to these well-described data, there is an inch of files and directories containing data but not described in the SM.pdf file. The PDB and Foce_Field directories contain important information but are not described in the SM.pdf file. What data the Definitions directory contains is probably only the authors know. Please organize the data, remove files or data that are unnecessary, not described or duplicated.
"The difference between the simulations is to be ascribed to different model constructions because most of the change in distance is spanned during the first 300 ns of simulation." (Linies 381-383) The reduction of fluctuations AND the stabilization of the distance between clusters at the end of the simulation is more a result of the transition of the studied system to equilibrium.
I do not know why the authors devote so much energy to describing the fluctuations of individual parts of the molecule during the dynamics. From the results of the prediction of the molecule structure, it is known that a significant part of the sequence is predicted with poor reliability (long unstructured loops), and some elements (the core of the protein) is predicted with greater precision and more structured. Thus, Figure 6 and the associated text show nothing new. The data is practically a reflection of the quality of the prediction what was already described earlier. The paper is very long-winded anyway suggests removing this part of the text (Lines - 242-374) and Figure 6.
The data and discussion on SASA (Table 5 and the text on it), would only make sense if the authors performed MD simulations for hydrogenase for which the experimental spatial structure is known and calculated the same parameter based on data from such a simulation. This would provide some kind of reference point for the experimental data and, at the same time, would be an important element in validating the quality of the obtained structural models, if the values obtained were similar. In order not to carry out such simulations, it would be possible to count these parameters taking only the crystallographic structure (the obtained value would be given without standard deviation).
Why Figure S2 in SM shows the starting structures for MD simulations. All analyses presented in the paper were performed for the last part of the trajectory. In Table 4 are given the differences between the start and end structures, it would be good to show the overlap of these structures (start and end )with each other.
I don't understand what the authors meant in the description of Figure 8 "Residues His 89-Leu 530 (H-domain) have been translated and rotated to minimize the RMSD of backbone heavy atoms between the three structures". This is the same amino acid sequence in all cases, so it would be easy to overlay the structures on top of each other to present them in the same orientation for ease of analysis. The drawing itself is ill-conceived, how is the reader supposed to know which Fe cluster is which ? How does the description of this figure relate to the data in Table 5. From the text, one can guess that for model 3, cluster H is more exposed to the solvent than in models 2 and 1 where the solvent penetration is reduced by domain F. However, if we look to Table 5, we have the same SASA values for model 2 and 3 and smaller values for model 1. These data do not combine into a unified and coherent whole.
For models other than model 1, did the authors identify water molecules that were close to Fe-d (Figure 5) ? If this did not happen for other models this is a very important premise to validate the quality of the models and reflect the structural features observed for other hydrogenases. What is missing is a deeper analysis regarding the mechanism of proton transport from the solvent to the Fe clusters, which is described in detail in Reference 37.
Lines 266 and 274 of the authors use the word "simulation" but this is about predictions and more specifically about structural models. Simulations using MD are described in the next section.
Line 303 is "figure shows" and should be "figure 5 shows."
The text needs intensive language correction. Authors use rather "unorthodox" or surprising words, e.g. "It can be observed that in simulation 3 the atoms of the [Fe4S4]H cluster are left accessible to water" (line 403). My guess is that it should be "cluster is moreaccessible to water than in the case of systems 1 and 2 ."
As I wrote above, the paper is too long, with the authors under different pages describing less important observations without focusing on critical validation of the obtained models using all available experimental data.
Comments on the Quality of English LanguageExtensive editing of English language required. I show example of bad wording, but quality of English is generally not good.
Round 3
Reviewer 2 Report (Previous Reviewer 3)
Comments and Suggestions for Authors
I have no more comments
Comments on the Quality of English LanguageNone
This manuscript is a resubmission of an earlier submission. The following is a list of the peer review reports and author responses from that submission.
Round 1
Reviewer 1 Report
Comments and Suggestions for Authors
The article “Predicting the structure of enzymes with metal cofactors: The example of [FeFe] hydrogenases” is devoted to the study of the structure of the enzyme [FeFe] hydrogenase from the microalgae Chlorella vulgaris 211/11P, which plays an important role in the production of molecular hydrogen. In the article, using modern molecular modeling methods, the structure of this protein was obtained (2 models). The dynamics of these models was studied using classical molecular dynamics. This paper is recommended for publication with major revision.
Line 11. Chorella vulgaris 211/11P
The typo. Correct it to Chlorella, please.
Lines 14-15 i) by a manipulation of the best known bacterial crystal 14
structure that includes iron-sulfur clusters.
This phrase in the abstract is not clear. It would be desirable to clarify what is meant by the word “manipulation”.
Lines 87-88. One 87
of the proposals of the algorithm is compared to a traditional homology model.
What were the similarities to homology modeling?
Lines 88-90. Interatomic 88
potentials are used to settle the initial structures in the environment similar to cell cytoplasm 89
and chloroplast intermembrane space. 90
What exactly was this similarity between the environment and the other and how was it ensured?
Line 152
as CX38CX2CX2C [24], while in Cvu we find CX50CX2CX2C. As shown by Nar proteins, 152
Please explain what the X with the number in the sequence means.
Besides FeS clusters, do cysteines form disulfide bridges between and within protein domains? Were they predicted and, if so, did the number of disulfide bridges differ between structures?
Lines 262-263. As for global 262
properties of the Hyd protein matrix, we first analyzed the gyration radius (Rg), the solvent 263
Lines 266-267. In the SM we describe in more details the protein matrix, where 266
the two simulations differ more significantly. 267
To account for global motions and differences in dynamics between the two protein models, was principal component analysis performed? Do the principal modes of the two proteins show similar or different dynamics?
Lines 268-275
As discussed in a recent work [27], active [FeFe] hydrogenases, with the exception 268
of Cr, show that the [4Fe4S]H-[4Fe4S]F distance (measured as the distance between the 269
centers of mass) is in the range 11-14 Å. In our case, we have two models in which the 270
distances between the two 4Fe4S clusters are in the range 12-20 Å (simulation 1) and 25-40 Å 271
(simulation 2). In Fig. 8 we report the behaviour of the distance between the 4Fe4S centers 272
of mass along both simulations. The comparison shows that simulation 1 better agrees 273
with previous observations. The difference between the two simulations is to be ascribed 274
to construction of the model. 275
Why was the statistical distance between clusters not taken into account when building the model? Is it possible that the increase in distance is caused by the initially incorrect location of the domains relative to each other?
Lines 307-308. Therefore, we argue that the F-domain interacts with the H-domain and 307
that depending on F-domain folding the H-domain is differently affected. This feature is a 308
This is an interesting observation. Have the dynamics of the H-domain been compared in the absence of the F-domain?
How different are the potential energies of domain interaction in both cases? What mechanisms lead to partial unfolding in the second case? It is recommended to supplement the article with data on the interaction of domains in both models.
Lines 309-311. Also, we can exclude that the F-domain has the 309
structure of CpI because this structure provides a non-functional Hyd, with the two clusters 310
at large distance one from each other. 311
Could the reason be the opposite - an incorrect prediction of the structure of the H-domain? Have you tried to derive the structure of this domain from homology modeling to compare with the Alpha Fold one?
Lines 408-409.
Summarizing, in this way we obtained two model structures with very similar H- 408
domain and completely different F-domain. 409
Is the model that was used to build the hybrid one the same as Model 1?
Figure S1. H- and F-domains are
colored according to Table 2 in main text.
I don't find the names of the colors in Table 2.
Lines 418-419
We inserted the protein (7957 atoms) in a simulation cell, adopting periodic boundary 418
conditions, with water and ions atoms to neutralize the system ([NaCl]=0.15 M). The whole 419
Please add the size of the periodic box.
How were the FeS clusters held in the protein matrix during the simulations?
Reviewer 2 Report
Comments and Suggestions for Authors
The manuscript "Predicting the structure of enzymes with metal cofactors: The example of [FeFe] hydrogenases" deals with structure prediction of FeFe hydrogenase from Chlorella vulgaris 211/11P strain. The structure of this protein is neither solved experimentally nor available in the AlphaFold DB of modeled protein structures. The authors predicted the structure fof the protein using AlphaFold, introduced the necessary FeS clusters into it, and performed additional molecular dynamics simulations.
The manuscript has some serious flaws, regarding the study design, data interpretation as well as presentation of the results.
The major problem with the presented data is that they are either based on incompletely investigated assumptions or based on an unreliable structure model.
The identification of the F-domain that contains additional FeS clusters: it is only described in text, but no data or sequence alignments for the N-terminus of the protein are given. The templates for the first 80 residues cannot be found in any homologous PDB structure (I checked it with sequence homology search servers HHpred and COMER). COMER found some SwissProt hits that are quite well aligned in the N-terminus, so probably the authors' suggestion is correct that some FeS binding residues are there in the N-terminus (Lines 149-157 in the text), but they have to be justified and illustrated by sequence alignment data in a similar fashion as the FeS binding residues are analyzed for the C-terminus of the protein. One BLAST alignment with one homolog (Fig. 3) is not enough, the assignment of the residues in Table 2 should be justified more.
Structure prediction: the N-terminus is not predicted by AlphaFold. The authors seem to ignore it (while the low pLDDT scores are clearly present in Fig. 4). Additional disoredered regions are also visible in the structure and are also ignored by authors. The same situation can be observed for homologs that have structures in AlphaFoldDB: the N-terminus is not well predicted in these models (I checked the SwissProt results from COMER search). There should be a reason why AlphaFold does not predict it, maybe non-standard settings or metagenomics-based MSAs could be useful to predict the structure for the N-terminus. However, it is clear that all the described studies regarding the first 80 residues of the protein, such as presented in sections 2.2 and 2.3, are invalid.
The authors also construct a "hybrid model". They take a part of PDB structure 6N59 and change the residues to be the same as in their protein. It is not a correct method to to homology modeling, because sequence homology of 6N59 region 119-206 is observed only in part, approximately from residue I-65 that corresponds to K-183 in the PDB structure 6N59. Thus the residues 1-65 are just assigned some arbitrary coordinates in this "hybrid model". The orientation of F-domain (N-terminus) in regard to the H-domain is also arbitrary in such a model. Furthermore, the quality of the resulting model is not justified by any statistical potential-based or other methods (ProSA, VoroMQA, QMEAN, etc.). As a result, I consider that this "hibrid model" is also unreliable, and the conclusions based on it are thus invalid.
The constructed models are not made available together with the manuscript, therefore the readers cannot see the structures, making the publication useless for the society.
The Supplementary Materials contain only a PDF file that gives a list of other files which should contain sequence alignments, settings for the software applications, etc. However, these files were not visible for me.
Other concerns:
* Giving the ID of the protein is not enough, the authors should specify also the database where the sequence could be found. I guessed that this is the NCBI protein sequences database, and found such an ID there. Then I also found a 100% identical protein in UniProt, but it is not good to make the readers guess the database.
* PDB entries should be cited both by PDB ID and corresponding article that describes the structure. Some images contain structures but it is not clear where are these structures from.
* Sequence alignment images should be probably better called Figures, not Tables.
* I did not understand how FeS clusters were modeled into the protein structure? AlphaFold models contain only standard aminoacid residues, not ligands or inorganic cofactors of the proteins.
* Not only pLDDT, but also PAE values of AlphaFold model should be analyzed to understand it's reliability.
* The molecular dynamics study is done using a partly unfolded protein structure. As I know, currently available MD methods still do not allow to fold protein structure.
* Line 35: maybe toxic CO and CN- ligands are not the natural ligands of this protein in a living organism?
* Line 110 states that all residues are observed in Cvu Hyd, but there are clearly differences from the consensus sequences in the presented alignment.
* Lines 134-135 state that the protein has enzymatic activity, however no references and no experiments are provided for that.
* Legend of Fig. 4: I think that DSSP is not used by AlphaFold.
* It would be nice to see more methods details in the main text of the article, it is not necessary to hide everything in the Supplementary Materials.
Conclusion:
I would suggest the authors to get back to sequence homology analysis and structure modeling part, produce a more reliable structure model, interpret the available protein sequence data more thoroughly, and only then turn to molecular dynamics and further studies. Now all the conclusions that are based on unreliable structure models are unreliable as well.
The English language quality is OK, but the manuscript has to be edited quite a lot to make it more comprehensible and correct all the mistakes present in the text.
Reviewer 3 Report
Comments and Suggestions for Authors
The paper by S.Botticelli et al. concerns the study of the spatial structure of enzymes of the hydrogenase family. I have the following comments on the content of the manuscript:
#1
The authors describe the annotation of the gene KAI3438965.1.and its associated amino acid sequence as a hydrogenase in a rather vague manner. Somewhat problematic is that the amino acid sequence associated with the KAI3438965.1. gene has already been classified as a hydrogenase. All publicly available databases describe this sequence as a hydrogenase, giving just a few examples:
UNIPROT database (www.uniprot.org) record E1ZR28
OrthoDB database (www.orthdb.org) record www.orthodb.org/?gene=554065_0:002121
NCBI CDD database (www.ncbi.nlm.nih.gov/Structure/cdd) classifies this sequence as belonging to domain architecture ID 10502698.
Thus, it seems that the research described by the authors on the identification of the amino acid sequence that is the product of the KAI3438965.1. gene was not preceded by an adequate review of the subject literature. Thus, the results described in Section 2.1 duplicate well-documented information available in open databases.
#2
Section 2.2 describes the construction of the model using the AlphaFold algorithm. It should be mentioned that the predicted structure is already deposited and available in the UNIPROT database (
AF-E1ZR28-F1). It is very difficult to follow the descriptions in this part of the paper. The authors refer to some structural elements that have not been previously described (e.g., during the description of the hydrogenase structures of Fig. 2).
#3
Section 2.3 describes the results of simulations of models of the hydrogenase under study using molecular dynamics. The description and presentation of the results is very chaotic and ill-conceived. However, the key problem is that the structure model derived from the AlphaFold algorithm (see Chapter 2.2) contains very long sections of sequences where the reliability of prediction is practically zero, and there are long loops in the model with unnaturally stretched conformations (see Figure 1S). The presence of these long stretches with undefined structure makes the simulation results quite questionable due to the poor quality of the starting structure. Not the main text but in the supplementary data, the authors state that during the simulation the structure of the molecule collapses due to the presence of the previously mentioned loops with unnatural conformation. The only possibly valuable result could have been an analysis of how the chosen force field parameters describe the dynamics of FeFe clusters.
In the reviewer's opinion, the paper largely lacks original results (Chapters 2.1 and 2.2). The results in Chapter 2.3 seem completely unreliable due to the very poor quality of the initial structural models. For the above reasons, I do not recommend the paper for publication.
Comments on the Quality of English LanguageNone